# Clay-Polymer Nanocomposites: Preparations and Utilization for Pollutants Removal

**DOI:** 10.3390/ma14061365

**Published:** 2021-03-11

**Authors:** Abdelfattah Amari, Fatimah Mohammed Alzahrani, Khadijah Mohammedsaleh Katubi, Norah Salem Alsaiari, Mohamed A. Tahoon, Faouzi Ben Rebah

**Affiliations:** 1Department of Chemical Engineering, College of Engineering, King Khalid University, Abha 61411, Saudi Arabia; 2Department of Chemical Engineering, Research Laboratory: Energy and Environment, National School of Engineers, Gabes University, Gabes 6072, Tunisia; 3Chemistry Department, College of Science, Princess Nourah bint Abdulrahman University, Riyadh 11671, Saudi Arabia; fmalzahrani@pnu.edu.sa (F.M.A.); kmkatubi@pnu.edu.sa (K.M.K.); 4Department of Chemistry, College of Science, King Khalid University, P.O. Box 9004, Abha 61413, Saudi Arabia; tahooon_87@yahoo.com; 5Chemistry Department, Faculty of Science, Mansoura University, Mansoura 35516, Egypt; 6Higher Institute of Biotechnology of Sfax (ISBS), Sfax University, P.O. Box 263, Sfax 3000, Tunisia

**Keywords:** water treatment, nanomaterials, adsorbents, clay-polymers nanocomposites

## Abstract

Nowadays, people over the world face severe water scarcity despite the presence of several water sources. Adsorption is considered as the most efficient technique for the treatment of water containing biological, organic, and inorganic contaminants. For this purpose, materials from various origins (clay minerals, modified clays, zeolites, activated carbon, polymeric resins, etc.) have been considered as adsorbent for contaminants. Despite their cheapness and valuable properties, the use of clay minerals as adsorbent for wastewater treatment is limited due to many factors (low surface area, regeneration, and recovery limit, etc.). However, clay mineral can be used to enhance the performance of polymeric materials. The combination of clay minerals and polymers produces clay-polymers nanocomposites (CPNs) with advanced properties useful for pollutants removal. CPNs received a lot of attention for their efficient removal rate of various organic and inorganic contaminants via flocculation and adsorption ability. Three main classes of CPNs were developed (exfoliated nanocomposites (NCs), intercalated nanocomposites, and phase-separated microcomposites). The improved materials can be explored as novel and cost-effective adsorbents for the removal of organic and inorganic pollutants from water/wastewater. The literature reported the ability of CPNs to remove various pollutants such as bacteria, metals, phenol, tannic acid, pesticides, dyes, etc. CPNs showed higher adsorption capacity and efficient water treatment compared to the individual components. Moreover, CPNs offered better regeneration than clay materials. The present paper summarizes the different types of clay-polymers nanocomposites and their effective removal of different contaminants from water. Based on various criteria, CPNs future as promising adsorbent for water treatment is discussed.

## 1. Introduction

Clean water is a crucial factor of our world and shows a vital role to perform all aspects of life and continuous development [1,2]. Despite the presence of several water sources on earth, clean drinking water resources are limited in many regions of the world. This fact is observed in big cities characterized by large dense populations, and quick industrialization. Related to the quick industrialization, scientists reported the existence of more than 700 carcinogenic and highly toxic inorganic and organic micro-contaminants. They are considered as persistent environmental pollutants non-biotransformable or non-biodegradable. Toxic inorganic metals included for instance chromium, mercury, cadmium, lead, arsenic etc. while organic contaminants included drugs, phenols, plasticizers, polybrominated diphenyl ethers, polychlorinated biphenyls, polynuclear aromatic hydrocarbons, and pesticides. 

Research efforts have been conducted to remove these contaminants from water/wastewater using several methods such as adsorption [3,4,5], electrolysis [6], electrodialysis [7], ion-exchange [8], reverse osmosis [9], conventional coagulation [10], and chemical precipitation [11]. Many of these methods, such as electrodialysis, electrolysis, ion-exchange, and reverse osmosis, are expensive and cannot be applied in developing countries. Conventional coagulation methods and chemical precipitation cause secondary contaminants requiring an additional treatment and increasing the treatment cost. Interestingly, adsorption is the most attractive process in developing countries due to its lower cost and its high efficiency to remove different types of contaminants. Generally, the selection of the useful adsorbent for water treatments is controlled by many factors such as the adsorption capacity of the material toward the target contaminant, cost/efficiency ratio, and the type and concentration of the contaminants present in water. The selected adsorbent must be available, non-expensive, easily regenerable, and have high selectivity toward target contaminants. Interestingly, the adsorption method produces low quantities of sludge, which is largely produced by other expensive methods like chemical precipitation. Efficient adsorbents from biological, organic or mineral origin have been developed for water treatment. The most important used adsorbents are polymeric resins [12], biomass [13], agricultural wastes [14], industrial by-products, silica beads [15], clay minerals, modified clays [16], zeolites [17], and activated carbon [18]. Among the cited materials, polymeric resins have the ability to adsorb a wide variety of organic, high surface area, wide pore structure range, and can be regenerated with no loss in their adsorption capacities. However, they have some disadvantages mainly the sensitivity to particle size, the poor water dissolution, and the pH dependence. Their performances are also affected by the specific surface area and the porosity, the type of the used resin and their non-suitability for all aromatic pollutants. Furthermore, they are not suitable for water treatment in developing countries due to their high cost of application.

Due to their cheapness and availability compared to other materials, clay minerals have been considered as adsorbent for the removal of several organic and inorganic contaminants from water [19]. However, their efficiency remains lower compared to activated carbon and zeolites. Generally, the practical use of clay materials as adsorbents is limited due to many factors such as the smaller surface area, the low adsorption tendency toward organic species, the difficulty in recovering clay particles from aqueous solutions, and their reduction after regeneration. In order to overcome these limitations, researchers have developed clay-polymer nanocomposites (CPNs) combining the advantageous characteristics of both clay minerals (cheapness, availability, eco-friendly, large surface area and stability) and polymers (mainly the high adsorption efficiency, high surface area and better regeneration) in a new adsorbent, which is nano-scaled or micro-scaled according to their modification process [20,21]. Interestingly, CPNs showed advanced properties (mechanical strength, low gas permeability and heat resistance) allowing cost reduction and enhancing their efficiencies in removing various contaminants from water. In addition to that, many of these CPNs can be synthesized from green materials, which are considered sustainable [22]. Consequently, it is very important to collect, analyze, and deepen the CPNs research work in order to develop safe and eco-friendly adsorbents applicable at a large scale for water treatment.

The present paper provides recent data concerning CPNs that have been prepared and used as adsorbents for the removal of pollutants from water/wastewater. All the utilities of CPNs in wastewater treatment will be discussed to clarify the future direction of the research work. This will help promote the development of a strategy for the design of CPNs adsorbents useful and applicable in water treatment process.

## 2. Structures and Types of Clay Minerals 

Clay minerals are an assembly of crystalline minerals. This structure consists of thin sheets (phyllosilicates) arranged in layers. Layers are composed of octahedral ([AlO_3_(OH)_3_]^6−^) and tetrahedral ([SiO_4_]^4−^) sheets connected through sharing of apical atoms of oxygen. Clay minerals layers stack via Van der Waals attractions to form interlayer. Clay minerals are classified into seven classes (Table 1): (i) fibrous-layered silicates such as sepiolite and palygorskite, (ii) 1:1 ratio of [AlO_3_(OH)_3_]^6−^ octahedral and [SiO_4_]^4−^ tetrahedral sheets called kaolinite class such as serpentine, halloysite, and kaolinite, (iii) 2:1 non-expanding class in which the ratio between [SiO_4_]^4−^ tetrahedral and [AlO_3_(OH)_3_]^6−^ octahedral sheets is 2:1 such as illite and mica, (iv) 2:1 uncharged class such as talc and pyrophyllite, (v) strongly expanding 2:1 class such as montmorillonite, (vi) limited expanding 2:1 class such as vermiculite, (vii) 2:1:1 class in which there is an extra brucite octahedral sheet sandwiched to form a 2:1 composition [23,24]. 

These clay minerals carry positive and negative charges due to isomorphous substitution of octahedral and tetrahedral sheets. Matrix and support of clay-polymers nanocomposites are provided by 2:1 class silicates like saponite, hectorite, and montmorillonite as well as palygorskite. These types of clay minerals are nano-scaled either in fiber diameter or in sheet thickness. Interestingly, organic compounds can be used for the modification of clay minerals to produce organoclays and this represents the first step of clay-polymer nanocomposites synthesis. Natural clay minerals are characterized by its hydrophilicity that allowed their saturation with lithium and sodium cations. Also, this hydrophilicity makes clay minerals incompatible with non-polar polymers but compatible with hydrophilic polymers like polyvinyl alcohol and polyethylene oxide [23]. Thus, non-polar polymers were made compatible with hydrophilic clay minerals via the synthesis of organoclays as preliminary stage. The conventional method of organoclays preparation is the combination of sulfonium, phosphonium, and ammonium with clay minerals [25]. Ion-exchange reaction is used to replace clay minerals cations by positively charged surfactant like hexadecyl trimethylammonium to make clay hydrophobic and allow its modification with hydrophobic polymers. Polar-nonpolar partition mechanisms provided to natural clay minerals via organoclays are known to improve the adsorption capacity of pollutants. Also, they allow an intercalation of pollutants and an increase of positive charges over clay minerals surfaces, which improve the chelation power [26].

Because of their cheapness, availability, and valuable properties, clay minerals are of large interest for their potential applications in various areas such as wastewater treatment. However, the application of clay minerals is limited due to many factors (low surface area, regeneration, and recovery limit, etc.) [19,22].

## 3. Nanocomposites of Clay-Polymers

Clay minerals can be used to enhance the performance of polymeric materials. The combination of clay minerals and polymers is considered as promising reinforcements to produce clay-polymer nanocomposites with advanced properties useful for pollutants removal. There are three main classes of nanocomposite clay-polymers; (i) exfoliated nanocomposites, (ii) intercalated nanocomposites, and (iii) phase-separated microcomposites [26]. In the exfoliated nanocomposites, the clay layers are fully dispersed within the polymer matrix. They are composed of ~1 nm clay layers separated within the matrix of polymer. In intercalated nanocomposites, the silicate crystalline layers are inserted within the polymeric chains to form nano-scaled clay-polymer composite. The phase-separated microcomposite, resulted from embedded polymer inside silicate layers, have similar properties as conventional microcomposites. This polymer nanocomposites display exceptional properties regarding the biodegradability, self-extinguishing behavior, heat distortion, flexural properties, tensile strength, and high modulus. CPNs may include many polymers in their fabrication like chitosan (CS), polystyrene, polypropylene, polyesters, polyurethanes, epoxies, and polyvinyl chloride (PVC). 

### 3.1. Clay-Polymers Magnetic Nanocomposites

Polymer matrix of CPNs could be filled with magnetic nanoparticles (NPs) to develop their functionality. For instance, magnetically coated nanoparticles were prepared by casting mix of oleic-acid-coated nanoparticles, poly(butyl acrylate) emulsion, and clay (laponite) [27]. However, this nanocomposite with low content of oleic acid exhibits greater saturation magnetization than that of pure oleic acid-coated magnetic nanoparticles component. This magnetic composite was synthesized via three steps; first, clay dispersion and mixed polymer were transformed into gel. Then, oleic acid-magnetic NPs and polymer were separately fixed in clay platelets and finally, all materials were merged in the main composite. Thus, polymeric matrix is used to fix oleic acid magnetic NPs and clay to produce exclusive nanomaterial. In another work, palygorskite–iron-oxide magnetic nanocomposite was synthesized via environmentally safe and economical precipitation synthesis [28]. 

### 3.2. Clay-Biopolymers Nanocomposites

Biopolymers (polynucleotides, nucleotides, polypeptides, polysaccharides, and proteins, etc.) are green and biocompatible materials useful in the synthesis of clay NCs with improved properties like functionality and reactivity. The improved properties offered by biopolymers allowed the application of the clay-biopolymers nanocomposites in different fields like packaging materials, tissue engineering, drug delivery, controlled pesticide, and electrochemical sensing. Chitosan is an example of biopolymers extracted from crustacean seafood wastes [29] and composed of units of *N*-acetylglucosamine and glucosamine. This biopolymer was used for the synthesis of clay nanocomposite useful for the removal of pollutants from water [30]. The synthesis of clay-chitosan nanocomposite was described in another work [31] as shown in Figure 1. 

### 3.3. Nanoclay of Polyacrylamide-Polymers Nanocomposites

Bentonite nanoclay polymer composite was synthesized using acrylic acid. First, alkali medium was used for the synthesis of acrylamide from acrylic acid. Then, bentonite was added in nitrogen atmosphere and cross-linking reactions were carried out via *N,N*′-methylenebisacrylamide. After that, the polymerization reaction was carried out at 70 °C in the presence of (NH_4_)S_2_O_8_ in order to produce the nanoclay-polymer [32]. The characterization of the obtained composite using various methods (Fourier transform infrared spectroscopy, scanning electron microscopy, and X-ray diffraction) showed a complete exfoliatation and dispersion of the bentonite layers in the composite after the polymerization, leading to an increase of the surface area. The new composite can be exploited in many fields such as agriculture [32]. 

### 3.4. Nanocomposites of Clay-Polysiloxane

Organoclays were usually used as matrix during the synthesis of NCs silicate-polysiloxane in the presence of little amount of H_2_O. Silanol terminated poly(dimethylsiloxane) was mixed with organo- montmorillonite, which is a mix of dimethyl ditallowammonium alkyl chains of C12, C14, C16, and C18 and cross-linked with tetraethylorthosilicate in the presence of catalyst to form NCs of silicate-polysiloxane [33]. The production of exfoliated NCs needs water and the mixing of the modified clay with silanol-terminated poly(dimethylsiloxane) should be carried out under sonication. Moreover, the nature of the clay modifier and silicon affects considerably the NCs synthesis. Hence, the absence of exfoliation and intercalation when montmorillonite is modified with benzyl dimethyloctadecylammonium cation was reported. Also, the use of silanol-terminated poly(dimethylsiloxane) containing only 14% (mol/mol) of diphenylsiloxane units showed only intercalation. Therefore, the optimization of the layered silicate exfoliation required right fitting between the matrix and organoclay [33]. 

### 3.5. Nanocomposites of Clay-Polystyrene

Polystyrene-clay composites were produced using the addition polymerization technique [34]. A vinyl monomer-montmorillonite intercalate was synthesized by the cation exchange process. Then, free-radical polymerization of styrene in the presence of vinyl monomer-montmorillonite intercalate was achieved leading to the production of polystyrene-montmorillonite materials. The solvation of vinyl monomer-montmorillonite intercalate using acetonitrile enhance the intercalation of styrene between the vinyl monomer- montmorillonite [34]. According to Vaia and Giannelis (1997), the melt intercalation is the best efficient method for the intercalation of polymer into organoclay [35]. The intercalation of polymers depends on the length of the alkyl chain in alkylammonium-exchanged smectites and on the annealing temperature. Under 160 °C, no intercalation was reported for chain having less than 12 carbon atoms. At 180 °C, intercalation occurred independently of the carbon atom number. However, it was reported that the intercalation was not affected by the molecular weight of the polystyrene [35]. 

### 3.6. Blends of Organoclay-Polymer

Organoclay with polymer blends such as poly(l-lactide), poly(ε-caprolactone)–clay, and poly-ethylene oxide–clay blends were prepared as reported by Ogata et al., (1997) [36]. Distearyl dimethylammonium-modified montmorillonite was mixed and blended with pellets of poly(l-lactide) in warmed chloroform. A 100-mm thick film was formed after evaporation of chloroform. Tactoids of unmodified montmorillonite were aligned on the surface of the film while intercalation of poly(l-lactide) into modified montmorillonite was absent [36]. The modified clay produced an interesting geometric superstructure (tactoids with several silicate monolayers) showing the enhancement of the Young’s modulus of the poly(l-lactide)-based composites. Also, storage and loss moduli were enhanced [36].

### 3.7. Hybrid Clay-Polyimide

Clay-polyimide NCs were synthesized by polymerization using pyromellitic dianhydride and 4,4′-diaminodiphenyl ether in dimethylacetamide solution and in the presence of organoclay [37]. The synthesized polyimide–clay nanocomposites with limit fraction of organoclay (2% wt/wt) showed remarkable reduction in the permeability of several gases (H_2_O, O_2_, CO_2_, etc.). Moreover, the increase in ratio of the clay mineral caused a decrease of thermal expansion coefficient and montmorillonite exhibited great exfoliation when NCs were synthesized [37].

### 3.8. Nanocomposites of Clay-Epoxy

To improve the solvent resistance and thermal stability of NCs, highly expanded onium-modified clay minerals were diffused into epoxide [38]. Epoxy support was formed using catalytic curing agents, anhydride, aromatic amines, and aliphatic amines having wide glass-transition temperature. Interestingly, new type of exfoliated structure was observed and the matrix tensile properties were improved by the reinforcement offered by the silicate nanolayers. The tensile properties of the resulted nanocomposites (epoxy−magadiite elastomeric) are comparable to other smectite clays [38].

### 3.9. Nanocomposites of Clay-Polyurethane

The solvation process allowed the intercalation of polyols into onium modified organoclays. Structure and length of onium ions greatly affected the intercalation process. The polymerization process of polyolisocyanate precursor–organoclay allowed the production of NCs with clay phase (50 Å basal spaces) intercalated in the cross-linked polyurethane network. An interesting benefit was observed with montmorillonites exchanged with chain onium ions containing a number of carbon atoms more than 12. The produced NCs showed higher toughness and strength compared to pristine polymer NCs [39]. 

Here, we reported the techniques used to develop nanocomposites of clay-polymers. Using clay minerals, significant technical advantages were achieved (mechanical, thermal, degradative, rheological properties, etc.) allowing the improvement of the new materials. The improved materials can be explored as novel and cost-effective adsorbents for the removal of organic and inorganic pollutants from water/wastewater.

## 4. Clay-Polymers Nanocomposites for Pollutants Removal

Based on their improved performances, CPNs showed the ability to remove various pollutants (bacteria, metals, phenol, tannic acid, pesticides, dyes, etc.) from water/wastewater. Below, the pollutant removals reported in the literature will be discussed. 

### 4.1. Biological Pollutants Removal

Drinking water containing microbes, such as *Entamoeba histolytica*, *Shigella sp.,* and *Salmonella typhi,* is responsible for dangerous water borne infections such as diarrhea, dysentery, and typhoid [40]. The reduction of pathogen population was traditionally ensured by the chlorination technology. However, nowadays, this technology becomes less used due the existence of soluble organic pollutants that allowed the formation of secondary toxic contaminants (chlorophenols, haloacetic acids, trihalomethanes, etc.) [41]. Also, chloramination process showed a higher potential to form nitrosamines than chlorination. Earlier, the use of sand slow filtration was suggested but with limited efficiency due to high cost of filter repair and blocking [42]. Interestingly, composites of clay-polymers were used for antimicrobial activity but the used polymers must be soluble. For instance, antimicrobial efficiency of copper-doped montmorillonite low-density polyethylene (LDPE) nanocomposites toward *E. coli* was examined [43]. The prepared CPNs contain montmorillonite-Cu^+2^ complex allowed high antimicrobial activity against *E. coli* (99%). Cell death was suggested via complexation between microbial nucleic acids and proteins with copper ions [44]. 

Similarly, other CPNs such as montmorillonite- polydimethyloxane-chlorhexidine acetate, and clay-polydimethyloxane-chitosan-silver showed the ability to retard the growth of different bacterial strains (*E. coli, Candida albicans, Pseudomonas aeruginosa, and Staphylococcus aureus*) [45,46,47,48]. *E. coli* was also removed via bentonite clay composite modified with starch-grafted quaternary ammonium ethers. The bacterial removal was achieved due to the presence of cationic monomers on clay surface [47]. The existence of monomer enhanced the composite with positive charges, that electrostatically attracted the negative phospholipids of the bacterial cell membrane causing cell disruption. Thus, CPNs are more efficient than polymers alone for bacterial removals. Also, chitosan-based NCs are known to have an antimicrobial activity. Interestingly, montmorillonite-chitosan NCs was more efficient for the removal of *S. aureus* and *E. coli* than montmorillonite and chitosan alone [49]. Additionally, bentonite silver and zinc oxide NCs were examined for bacterial removal via adhesion and killing mechanism [50]. Furthermore, *E. coli* was removed via NCs of rectorite modified with PVA, chitosan, and sodium-dodecylsulfonate compared to the single components [46,51]. High antimicrobial activity was also reached via chitosan-organoclay and chitosan-montmorillonite modified with Ag^+^ nanoparticles [52,53].

Generally, the microbial removals using CPNs is controlled by various factors such as the properties of the adsorbent (surface charge, composition, etc.), the characteristics of the bacterial strains and the operating conditions (pH, temperature, dosage, etc.). In this context, most chitosan NCs showed antibacterial activity in acidic medium but, *N*-(2-hydroxyl)propyl-3-trimethyl ammonium chitosan Cl displayed antibacterial activity in a large range of pH [46,54,55]. Finally, the reuse of any CPNs is very important due to the cost of application. As reported in the literature, the reuse of CPNs applied for microbial removal was achieved using various reagents such as NaClO, HCl, and steam [47,56]. As a result of their high antimicrobial activity, CPNs can be applied efficiently to disinfect contaminated water.

### 4.2. Organic Pollutants Removal 

The presence of dangerous toxic organic pollutants (organic acids, phenols, pesticides, dyes, etc.) may cause environmental problems even at trace amounts. Interestingly, these organic contaminants were removed using CPNs as adsorbent, due to the existence of polymeric hydrophobic parts on the NCs surfaces [57,58,59,60,61,62,63,64,65,66,67,68,69,70,71,72,73,74,75,76,77,78,79,80,81,82,83,84,85,86,87,88,89,90]. In this perspective, several CPNs in the form of powder or tablets were prepared using the thin-coated layer methods. These methods provide an enhancement of the adsorbent reusability after adsorption of toxic contaminants [57]. As listed in Table 2, different organic contaminants can be removed from water via adsorption process using CPNs.

#### 4.2.1. Phenol, Tannic Acid and Pesticides Removal

The direct release of organic pollutants in water greatly affects the environment. Among these organics, polyphenolic tannic acid, which resulted from decomposed organic materials, is highly harmful for organisms in aquatic environment. Commonly, tannic acid compounds exist in surface water and soil. Consequently, it is important to find the proper way for their removal. In this context, attapulgite clay coated with chitosan was used as a new adsorbent for tannic acid removal with an adsorption capacity of 95.3 mg/g. This adsorption resulted from Van der Waals attractions, H-bonding, and electrostatic interactions [58]. Interestingly, this nanocomposite was improved to adsorb more tannic acid (456 mg/g) using solid matrix of protonated palygorskite clay (20% wt/wt) with chitosan resin microspheres [59]. In this study, the reusability of this novel nanocomposite was studied using hydrochloric acid (0.1 M) as an eluent up to three cycles showing a significant decrease of the adsorption capacity (58 mg/g) of tannic acid [59].

Similarly, phenolic materials such as 4-chlorophenol and phenols were removed using alginate polymer support incorporated by clay modified with a surfactant hexadecyl trimethyl ammonium. This adsorbent allowed a removal capacity of 0.119 mg/g and 0.335 mg/g respectively for 4-chlorophenol and phenols [60]. Also, trichlorophenol and trinitrophenol were removed using nano- montmorillonite synthesized from poly-4-vinylpyridine-co-styrene with removal efficiency of 60% and 99.6%, respectively. In this study, it has been suggested that a weak Van der Waals interaction is involved in the adsorption mechanism [61]. In another study, phenolic materials were removed by nanocomposite prepared using cetyltrimethylammonium poly(diallyldimethylammonium) to modify montmorillonite clay. Although the surfactant was not correctly intercalated inside the clay montmorillonite, the adsorption capacity toward phenols was enhanced [62].

Wastewater was also treated to remove pesticides like 2,4-dichlorophenol and atrazine using CPNs as adsorbents. 90% poly-4-vinylpyridine-co-styrene was used to modify montmorillonite allowing the removal of 99% of atrazine within only 40 min [63]. Interestingly, the synthesized NCs decreased the presence of pesticide to 3 μg/L during the use of fixed bed experiment. However, the adsorption capacity of these NCs was decreased in the presence of other organic soluble contaminants due to the competition occurred on the adsorbent surface. In another study, dinitrophenols were removed using membranes filled with NCs of organo- montmorillonite/ polyethersulfone. The NCs were prepared using wet-phase inversion and solution dispersion methods [64]. The addition of 4% of organo- montmorillonite enhanced the nitrophenols removal. Nitrophenols were removed via hydrogen bonding over NCs surfaces at pH 4.6. Furthermore, more than 70% of mecoprop and clopyralid anionic pesticides was removed over hexadimethrine/ montmorillonite NCs due to electrostatic interactions with adsorbent NH^4+^ groups [65]. 

#### 4.2.2. Dyes Removal 

The dyes removal from water/wastewater using clay/polymers nanocomposites was reported by many studies. In this context, many researchers reported the use of chitosan polymer [66,67,68,69,70,71,72,73]. For example, clay-polymer nanocomposite of chitosan/bentonite was synthesized and tested for the removal of azo dye tetrazine, known by its harmful effect on aquatic organisms [66]. According to this study, the pH value is the driving force of the adsorption process. At low pH, the +ve charge was produced allowing the electrostatic attractions between dye and nanocomposite surfaces. However, at higher pH value, the charge became –ve reducing the adsorption process due to the repulsion forces between NCs surfaces and the dye. The cross-linked chitosan in the nanocomposite with epichlorohydrin becomes water insoluble improving the adsorption process. Also, methylene blue (MB) dye was removed using chitosan/clay magnetic beads at pH 3–12 with weight ratio of clay to chitosan of more than 0.5. These magnetic NCs provided faster removal of MB within 13 min (50% removal efficiency and 83 mg/g maximum adsorption capacity) [67]. The adsorption process occurred via a mechanism of electrostatic attractions between clay –ve charge and dye +ve charge for wide pH range. Similarly, chitosan-clay NCs were used for the removal of Rhodamin-6 G with adsorption capacity of 446.43 mg/g [68]. At high pH, the adsorption of this positively charged dye occurred due to the attraction with the negatively charged adsorbent. The pseudo-second-order model was found to explain the adsorption kinetics of Rhodamin-6 G [68]. In the same way, high removal efficiency (99.99%) was obtained for MB using chitin-clay NCs with an adsorption capacity of 152.3 mg/g. The adsorption was greatly linked to the pH and followed the Langmuir isotherm [69]. Chitosan modified nano-organoclay was also applied for the removal of reactive red-141 (adsorption capacity of 440 mg/g) and reactive blue-21 (adsorption capacity of 477 mg/g). The adsorption data of the two dyes followed pseudo-second-order model [70]. For reactive dye (Yellow 3 RS), the adsorption capacity reached 71.39 mg/g over chitosan modified palygorskite clay NCs, which is significantly higher than that obtained with unmodified clay materials. (6.4 mg/g) [71]. At high pH, the adsorption was decreased due to the repulsion forces between similar charges while at low pH the positive charge was formed over chitosan surface due to the existence of amino groups allowing the adsorption via electrostatic attractions [71].

Other polymer types were used to formulate PNCs useful for dyes removal [74,75,76,77,78]. For example, reactive yellow K-4G, and disperse yellow-brown S-2RFL were removed via NCs of poly(epicholorohydrin-dimethylamine) and bentonite with maximum adsorption capacity of 30.9 mg/g and 12.5 mg/g, respectively. The complete dyes removal was reached via the optimization of polymers quantity [74]. Using the same approach, humic acid immobilized-amine-modified bentonite–polyacrylamide nanocomposite was tested for the removal of various dyes (malachite green, MB, and crystal violet) from wastewater [76]. This clay-polymer nanocomposite showed maximum adsorption capacity of malachite green, MB, and crystal violet of 658, 649, and 511 μmol/g, respectively at pH 4.8. In This study, intraparticle diffusion mechanism was suggested [76].

Recently, De Sousa et al. synthesized hybrid clay-polymer nanocomposite using sodium bentonite and ureasil-poly(ethylene oxide) via sol gel method for the removal of MB from water. Interestingly, MB was removed rapidly and efficiently through the adsorption process [77]. In another work, MB was removed from water using hybrid membranes based on kaolin and polystyrene. These membranes were fired at 1000 °C allowing the modification of polystyrene and the formation of cavities inside the clay matrix. The resulted materials have porous structure with expanded surface area. An optimum removal of MB from water was obtained with clay membrane loaded with 20% (wt/wt) polystyrene [78]. The mechanism of MB removal over clay-polymer composites was described in the literature [31] as shown in Figure 2.

Generally, dyes removal efficiency offered by the new CPNs varied depending on the polymerization process and the used raw materials. For each dye, the adsorption performance (removal rates and the adsorption capacity) varied from one experiment to another depending on the CPNs nature and operating conditions (dosage, pH, temperature, contact time, dye type, initial dye concentration, etc.).

### 4.3. Inorganic Pollutants Removal

#### 4.3.1. Toxic Gases Removal 

Clay-polymer nanocomposites can be used for the adsorption of toxic gases like ammonia and metallic gases such as mercury Hg^0^. For example, chitosan-bentonite composite was used as adsorbent for the elimination of Hg^0^. In this study, the authors found that the low surface area of composite compared to single bentonite caused less removal of Hg^0^ [91]. Also, a composite, synthesized from acid-activated bentonite and natural palygorskite via polymerization reaction between *N,N’*-methylenebisacrylamide and acrylic acid, was used for the adsorption of ammonia [92]. To increase the ability to remove ammonia, Lewis acid-base interaction was used to bind copper divalent ions with the composite. Among all acid-treated bentonite/acrylic acid polymer composites, 66% acid-treated bentonite exhibited the highest adsorption capacity. However, 75% palygorskite showed the maximum adsorption capacity among all palygorskite/acrylic acid polymer nanocomposites. The differences in adsorption capacity between acid-treated bentonite/acrylic acid polymer and palygorskite composites are related to the existence of mesopores (diameters between 2 and 50 nm) in the composite (copper-complexed clay/poly-acrylic acid composite), that allowed the interaction between polymers active sites and ammonia [92]. 

#### 4.3.2. Metalloids and Heavy Metals Removal

Urbanization, fertilizer application, and rapid industrialization are the major sources of water toxic metalloids and heavy metals such as As, U, Se, Sb, Cr, Cd, Ni, Pb, and Cu. Metalloids and heavy metals represent a serious danger for human and living organisms. These pollutants can be accumulated inside different organisms causing harmful effects. Therefore, their removal from water is essential for healthy environment [93]. 

Efforts have been conducted to remove the toxic heavy metals from water using clay as the support and chitosan as the polymer [94,95,96,97,98,99,100,101,102,103,104,105,106,107,108,109,110,111]. Divalent copper ions were removed from water using chitosan–silver NPs clay composite with a maximum adsorption capacity (181.6 mg/g) at pH 7 [94]. The high adsorption at pH 7 resulted from the interaction between chitosan functional groups and Cu(II) ions. At high pH, the formed cupric hydroxide decreased the adsorption process of copper ions. Another nanocomposite, poly-methacrylic acid grafted chitosan–bentonite, was tested for the removal of Cd (II), Pb (II), and Hg (II) ions. In this case, removal values of 78%, 89%, and 94%, respectively for Cd (II), Pb (II), and Hg (II) were obtained at high pH [95]. For the same purpose, chitosan/attapulgite composites were synthesized and tested for the removal of Cr(III) and Fe(III) from aqueous solution. The maximum adsorption capacity was 11.65 and 10.41 mg/g for Cr(III) and Fe(III), respectively. These values were obtained with adsorbent dosage of 0.2 g/l. However, the adsorption efficiency was found to be increased by increasing the temperature. Generally, the metal sorption is controlled by the process of electrons sharing between adsorbent and metals or through the covalent bonding [96]. 

In the same context, Chitosan–Al-pillared montmorillonite nanocomposite was used to remove Cr (VI) with an adsorption capacity of 15.68 mg/g [97]. Lead and copper divalent ions were removed with a similar nanocomposite prepared with clay: chitosan ratio of 1:0.45 to remove Pb (II) and Cu (II) at pH 6.5 with an efficiency of 99.5% and 96%, respectively via chemisorption mechanism. This nanocomposite when treated with nitric acid showed excellent reuse results [98]. Also, clay-polymer nanocomposite was successfully synthesized using chitosan and fibrous clay minerals like palygorskite. Different mass ratios of palygorskite to chitosan (1:1, 2:1, and 1:2) was prepared and applied for the removal of Pb (II) from water. A maximum removal capacity (201.5 mg/g) was obtained using the ratio 1:1 [99]. The availability of pores equally from both clay and polymer is the reason for high removal of Pb (II). In another study, the lead adsorption was increased by many folds from aqueous solution when the clay mass increased by 10% with respect to the polymer polystyrene inside the composite [100]. In a recent research, high amount of lead ions were removed using ion-imprinted polymer/ montmorillonite NCs [101].

For the same purpose, chitosan-grafted polyacrylic acid bentonite was used to remove various metals (Ni (II), Cd (II), Zn (II), and Cu (II)) from water. Maximum removal was obtained at pH of 8, 6, 7, and 6, respectively for Ni (II), Cd (II), Zn (II), and Cu (II) [102]. The maximum adsorption at high pH can be explained by the existence of negative charge over the adsorbent surfaces allowing the chelation of positive heavy metals. Similarly, chemisorption mechanism explained the efficient metal removal (Ni (II), Cu (II), and Pb(II)) from aqueous solution by chitosan immobilized on bentonite using ethylene glycol diglycidyl ether as a cross-linker [103]. This adsorbent offered high adsorption capacity of 15.82, 21.55, and 26.38 mg/g respectively for Ni (II), Cu (II), and Pb(II) [104]. An exothermic adsorption was observed for these metals as concluded from the thermodynamic parameters with the decreasing of the entropy behavior [105,106,107,108,109,110]. Additionally, adsorption of Cu (II) was spontaneous only at 25 °C, while that of Pb (II) was always spontaneous. Concerning Ni (II), the adsorption was non-spontaneous at 25–55 °C [104]. In the same work, the desorption process was studied showing high recovery of Cu (II) (92%) obtained with HCl solution (pH 1) and under agitation (2 h) [104]. Remarkably, the effectiveness of membrane of chitosan-kaolin-based ceramic synthesized using PVA as the chelating material for the removal of Hg and As from aqueous solution was also demonstrated [111]. 

The interesting opportunity offered by the implication of chitosan in the adsorption was well performed. The mechanism of heavy metals removal over chitosan-clay nanocomposite was described in the literature [31] as shown in Figure 3.

On the other hand, various clay NCs involving other types of polymers were tested for the removal of metals. For example, montmorillonite modified with *N*-methyl-*D*-glucamine-based monomer through radical polymerization reaction to form water soluble nanocomposite was used with ion-exchange resin for the removal of As [112]. Interestingly, the resin removed 56 mg/g of arsenic via a Langmuir adsorption isotherm and decreased its level below the allowed limits according to the World Health Organization (WHO). Recently, in situ polymerization was used for the preparation of magnetic clay –polymer nanocomposite by using monomer methyl methacrylate, iron oxide nanoparticles, and bentonite [113]. This magnetic nanocomposite adsorbed 113 mg/g of Cr (VI) from water. Intraparticle diffusion equation was used to approve that the adsorption process was reached via film-diffusion. Another strategy was developed by Ravikumar and Udayakumar et al. 2020 using *M. oleivera* biopolymer and bentonite (mass ration 1:1) to produce green nanocomposite by the solution processing method. The application of this nanocomposite for the removal of Cd (II), Cr (II), and Pb (II) from aqueous solution via coagulo-adsorption process showed an average removal efficiency of heavy metals of 99.99% at pH 6–8, 2–4, and 5–7 for Cd, Cr, and Pb, respectively with a dosage of 5 g/L [114].

The reported data showed various possibilities useful to remove the metal from water as summarized in Table 3. Nevertheless, in some cases, the water can be contaminated by the nanocomposite itself, which should be taken into consideration and studied in more details. 

## 5. Factors Making Clay-Polymers Nanocomposites Promising Materials for Wastewater Treatment

This work reviews the most appropriate clay-polymers nanocomposites available for the treatment of water/wastewater containing different types of pollutants. Similar to other fields of fabricated hybrid composites, clay-polymers nanocomposites associate the advantages of their individual parts, like chemical stability, high surface charge, high surface structure, and mechanical stability as well as enhanced adsorption capacities to different pollutants that allow efficient water purification. Also, clay-polymers nanocomposites withstand the fixed bed systems high temperatures and pressures due to the improvement offered by polymers (thermal and the mechanical stability). Therefore, clay-polymers nanocomposites can be used in industrial scale with low exhaustion during long-term use [130]. Additionally, clay-polymers nanocomposites provide great price advantage compared to other famous adsorbents (zeolites, activated carbon, etc.) related to the use of both clay minerals and functional polymers. This cost advantage can be further enhanced with the reuse opportunity of the clay-polymers NCs that reduce the overall cost of treatment [131]. In addition to cost, clay-polymers nanocomposites can be applied at low dose allowing its potential use at industrial scale. 

Generally, unmodified clay minerals such as montmorillonite, bentonite, and kaolinite showed poor adsorption capacity for many pollutants as reported by Styszko et al., (2015) [132]. However, several CPNs exhibit higher adsorption capacities than mineral clays. For example, Cu (II) ions were removed from water with adsorption capacity of 44.9 mg/g using natural mineral clay while this value was folded more than twice for CPNs [133]. Similarly, maximum adsorption capacity of dyes was found for clay composites. In this context, raw kaolinite and montmorillonite showed dye adsorption capacity of 29 and 19 mg/g, respectively [134]. However, CPNs showed higher adsorption capacities as reported in Table 3. In addition to that, mineral clays adsorb better in acidic solutions making their application difficult for the treatment of water/wastewater having neutral pH [132].

As reported in Table 2 and Table 3, the pollutants removal efficiency allowed by synthesized CPNs varied depending on the polymerization process and the used raw materials. In some cases, the new nanocomposites exhibit an interesting removal capacity. For example, chitosan-g-poly (acrylic acid)/ montmorillonite showed an adsorption capacity of 1895.0 mg/g for methylene blue [72]. However, for poly(acrylic acid-co-2-acrylamido-2- methylpropanesulfonic acid)/montmorillonite, the adsorption capacity was only 215.0 mg/g for the same dye [85]. Result variability was also observed for other pollutants (heavy metals, bacteria, etc.). Therefore, the removal rates and the adsorption capacities varied depending on the CPNs nature and on the operating conditions (dosage, pollutants, pH, temperature, contact time, etc.). Many of these adsorbents have the ability to reduce the level of biological pollutants to the safe limits considered by WHO.

An additional advantage of clay polymer nanocomposites is their ability to be used in batch and fixed bed systems. When comparing both systems (batch and fixed bed), fixed bed system is most efficient for the continuous removal of contaminants and offered larger regeneration capacity of the adsorbent. Therefore, nanocomposites can be easily separated from reaction medium than unmodified mineral clays. This makes clay polymer nanocomposites more cost effective than other conventional adsorbents [135]. Moreover, clay polymer nanocomposites are more stable in water compared to their individual parts. 

The advantages of clay polymer nanocomposites can be summarized in the economic cost, the improved performances, and the easy synthesis in large quantities. Therefore, more researches are needed to produce the most promising clay polymer nanocomposites and evaluate the process efficiency at large scale for the removal of different contaminants (biological, organic, and inorganic) from water/wastewater. Furthermore, the future research of clay polymer nanocomposites can include the development of clay polymer NCs membranes. Membranes will provide effective treatment with low- cost embedded adsorbents. Also, study should be addressed to the synthesis of clay polymer NCs hydrogels, which may be considered more attractive for water/wastewater treatment due to their high capacity and ease of regeneration. Hydrogels are usually excluded from water purification due to poor strength and poor mechanical properties that can be overcome via the synthesis of clay polymers NCs hydrogels. Additionally, it will be promising to fabricate multi-functional clay polymer NCs having the ability to adsorb simultaneously different types of cationic and anionic contaminants.

## 6. Conclusions

This review article summarizes the different types of clay polymer nanocomposites and their effective use for the removal of pollutants from water/wastewater. The introduction of CPNs is induced by the advantages offered by clay minerals such as their cheapness, availability, and valuable properties. Therefore, the combination of clay minerals and polymers as promising reinforcements produce nanocomposites of clay-polymers with advanced properties (high capacity, simple synthesis, eco-friendly, cost advantages, etc.) useful for pollutants removal. CPNs may include various polymers in their fabrication such as chitosan, polystyrene, polypropylene, polyesters, polyurethanes, epoxies, and polyvinyl chloride, using appropriate synthesis techniques. Interestingly, the developed CPNs showed significant advantages of established adsorbents for water/wastewater treatment. The current literatures reported the removal of different pollutants (bacteria, metals, phenol, tannic acid, pesticides, dyes, etc.) using clay polymers nanocomposites showing them as promising materials for water/wastewater treatment. However, the majority of results were collected based on research experiments conducted for pollutants in aqueous solutions at laboratory scale. Therefore, further investigations are needed to evaluate the process efficiency using real wastewater at large scale. Moreover, it is very important to introduce sustainable waste materials such as agricultural and industrial wastes in the production of new CPNs-based adsorbents. Also, the competitive applicability of these innovative adsorbents should be evaluated taking into account the various parameters linked to the material proprieties (material degradation, alteration, life cycle, and regeneration) and to the generated wastes including the disposal of the loaded pollutants and the chemicals used for the process of adsorption/desorption. 

## Figures and Tables

**Figure 1 materials-14-01365-f001:**
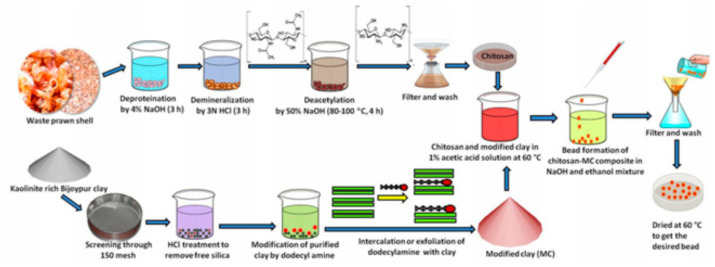
The scheme of clay-chitosan nanocomposite synthesis [31].

**Figure 2 materials-14-01365-f002:**
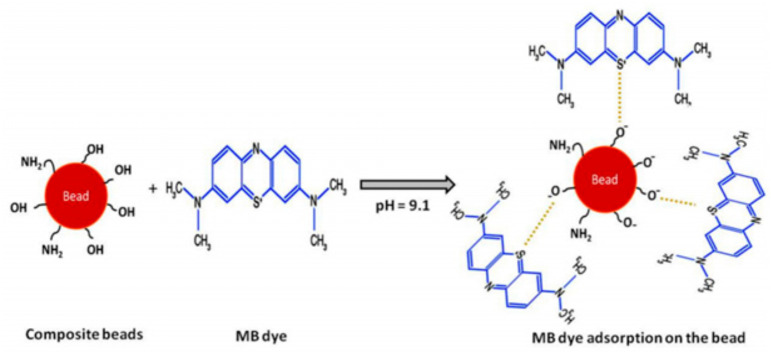
The mechanism of methylene blue (MB) removal over clay-polymer composite [31].

**Figure 3 materials-14-01365-f003:**
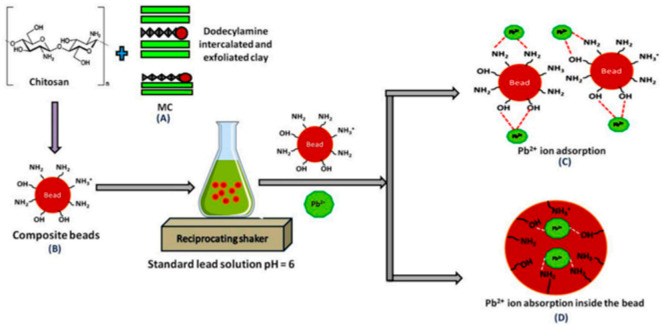
The mechanism of Pb^+2^ ions removal over chitosan-clay composite [31].

**Table 1 materials-14-01365-t001:** Classification of clay minerals [23,24].

Classes	Structure	Formulas
Kaolinite and serpentine	two-sheet phyllosilicates	kaolinite Al_4_[Si_4_O_10_](OH)_8_ serpentine Mg_6_[Si_4_O_10_](OH)_8_
Micas	three-sheet phyllosilicates	_2_Al_4_[(Si_>6_Al_<2_)O_20_](OH)_4_·nH_2_O
Vermiculite	expanding three sheet phyllosilicates	(Mg,Fe^2+^, Fe^3+^)_6_[(Si>Al)_8_O_20_](OH)_4_·nH_2_O
Smectites	strongly expanding three-sheet phyllosilicates	montmorillonite: M^+^_x+y_(Al,Fe^3+^)_4 − y_(Fe^2+^,Mg)_y_[Si_8 − x_Al_x_O_20_]·(OH)_4_·nH_2_O, beidellite: M^+^_x_Al_4_[Si_8 − x_Al_x_O_20_] (OH)_4_·nH_2_O, nontronite: M^+^_x_Fe^3+^[Si_8 − x_Al_x_O_20_] (OH)_4_·nH_2_O, saponite: M^+^_x_Mg_6_[Si_8 − x_Al_x_O_20_] (OH)_4_·nH_2_O.
Pyrophyllite and talc	nonswelling three-sheet phyllosilicates	pyrophyllite: Al_4_[Si_8_O_20_] (OH)_4_ talc: Mg_6_[Si_8_O_20_] (OH)_4_
chlorites	four-sheet silicates	Al_4_[Si_8_O_20_] (OH)_4_Al_4_(OH)_12_
Palygorskite and sepiolite	sheet-fibrous structure	palygorskite: Mg_5_[Si_8_O_20_] (OH)_2_(OH_2_)_4_·4H_2_O sepiolite: Mg_8_[Si_12_O_30_] (OH)_4_(OH_2_)_4_·nH_2_O

M^+^: represents adsorbed alkali cations (Na^+^).

**Table 2 materials-14-01365-t002:** Removal of different organic contaminants from water using clay-polymers nanocomposites (CPNs) adsorbents.

Adsorbent	Adsorbate	Temp. (^o^C)	pH	Removal Efficiency (%) or Adsorption Capacity (mg/g)	Isotherm Model	Kinetics Model	Ref.
Chitosan-coated attapulgite	Tannic acid	–	5.5	95.2 mg/g	Freundlich	Pseudo-second order	[58]
palygorskite/chitosan resin microspheres	Tannic acid	–	8.0	455.0 mg/g	Langmuir	Pseudo-second order	[59]
Chitosan/bentonite	Tartrazine	47	2.5	294.1 mg/g	Langmuir	Pseudo-second order	[66]
Chitosan-g-poly (acrylic acid)/montmorillonite	Methylene blue	–	6.5	1895.0 mg/g	Langmuir	Pseudo-second order	[72]
Chitosan/montmorillonite	Congo red	30	4.0	–	Langmuir	Pseudo-second order	[73]
Amino-modified polyacrylamide–bentonite NCs	Malachite green	30	6.0	656.5 mg/g	Freundlich	Pseudo-second order	[78]
Mixture of bentonite, acrylic polymer, and polyethylene-diamine (Zwitterionic adsorbent)	Acid Red and Brilliant Green	27	–	70.09 and 255.99 mg/g	–	–	[79]
AAm-AMPSNa/clay hydrogel nanocomposite and acrylamide (AAm)-2-acrylamide-2-methylpropanesulfonic acid sodium salt (AMPSNa) hydrogel	Brilliant cresyl blue and Safranine-T	25	–	494.2 and 484.2 mg/g	Langmuir	Pseudo-second order	[80]
Humic acid-modified bentonite	2,4-dichlorophenol	30	6.5	14.23 mg/g	–	–	[81]
montmorillonite/layer double hydroxide	Methyl orange	–	–	88%	–	–	[82]
montmorillonite /layer double hydroxide	Methylene blue	–	–	74%	–	–	[82]
Chitosan/bentonite	Amido Black 10B	20	2.0	323.6 mg/g	Langmuir	Pseudo-second order	[83]
Hydrogels of Kappa-carrageenan-g-poly(acrylamide)/sepiolite NCs	Crystal violet	Ambient Temp.	10.0	47.0 mg/g	Langmuir	Pseudo-second order	[84]
Poly(acrylic acid-co-2-acrylamido-2-methylpropanesulfonic acid)/ montmorillonite	Methylene blue	25	10.0	215.0 mg/g	Redlich–Peterson	Pseudo-second order	[85]
Alginate–clay quasi-cryogel beads	Methylene blue	40	–	181.8 mg/g	Langmuir	Pseudo-second order	[86]
Chitosan/bentonite	Malachite green	37	6.0	435.0 mg/g	Langmuir	Pseudo-second order	[87]
polyaniline/montmorillonite clay nanocomposites	Green 25	20	6.0	100%	Langmuir	Pseudo-second order	[88]
Tetraethoxysilane-functionalized Na-bentonite (2 wt%) incorporated into polysulfone/polyethylenimine membranes	Methylene blue	Room Temp.	–	98.9%	–	–	[89]
carboxy methyl cellulose/ nano-organobentonites	Nine pesticides	–	–	57–100%	–	–	[89]

**Table 3 materials-14-01365-t003:** Water treatment from different inorganic contaminants over clay-polymers nanocomposites (NCs) adsorbents.

Adsorbent	Adsorbate	Temp. (°C)	pH	Removal Efficiency (%) or Adsorption Capacity (mg/g)	Isotherm Model	Kinetics Model	Ref.
Bentonite/humic acid	Cu (II)	30	6.50	22.41 mg/g	–	–	[81]
Chitosan/attapulgite	Cr (III)	45	5.0	65.37 mg/g	Langmuir	Intraparticle diffusion	[96]
chitosan grafted poly acrylic acid bentonite composites	Cd (II)	25	6.0	51.60 mg/g	Langmuir	–	[96]
Chitosan/attapulgite	Fe (III)	45	3.0	62.51 mg/g	Langmuir	Intraparticle diffusion	[96]
Chitosan–Al-pillared MMT nanocomposite	Cr (VI)	25	6.38	15.68 mg/g	Langmuir	Pseudo-second order	[97]
Chitosan–Al-pillared montmorillonite NCs	Pb (II)	25	6.5	99.6%	Freundlich	Pseudo-second order	[98]
Chitosan grafted poly acrylic acid bentonite composites	Ni (II)	25	7.0	49 mg/g	Langmuir	–	[102]
Chitosan grafted poly acrylic acid bentonite composites	Cu (II)	25	6.0	88.60 mg/g	Langmuir	–	[102]
Chitosan immobilized on bentonite	Pb (II)	35	–	26.39 mg/g	Freundlich	Pseudo-second order	[104]
Polyaniline modified bentonite	U(VI)	20	6.5	14.10 mg/g	Langmuir	Pseudo-second order	[115]
Bentonite/thiourea-formaldehyde composite	Mn (VII)	–	4.0	14.82 mg/g	Langmuir	Pseudo-second order	[116]
Alginate–montmorillonite nanocomposite	Pb (II)	–	6.0	244.7 mg/g	–	–	[117]
Alginate–montmorillonite nanocomposite	Mn (II), Fe (III), Ni (II), Zn(II)	–	6.0	100%	–	–	[117]
Chitin/bentonite nanocomposite	Cr (VI)	–	4.0	443.72 mg/g	Freundlich	–	[118]
Na-montmorillonite /cellulose	Cr (VI)	–	3.8–5.5	22.3 mg/g	Langmuir	Pseudo-second order	[119]
Chitosan/PVA/bentonite nanocomposite	Hg (II)	–	–	360.74 mg/g	–	–	[120]
Chitosan and montmorillonite	Se (VI)	–	–	18.50 mg/g	–	–	[121]
Poly(methacrylic acid) grafted chitosan/bentonite grafted chitosan /bentonite	Th (IV)	30	5.0	110.60 mg/g	Langmuir	Pseudo-second order	[122]
L-cysteine modified bentonite-cellulose nanocomposite	Cu (II)	50	–	32.37 mg/g	Langmuir	Pseudo-second order	[123]
Poly(acrylic acid-co-acrylamide)/attapulgite	Cu (II)	–	6.0	69.76 mg/g	–	–	[124]
Cellulose-graft- polyacrylamide/hydroxyapatite	Cu (II)	–	7.0	176.0 mg/g	–	Pseudo-second order	[125]
Chitosan/clinoptilolite	Ni (II)	25	5.0	247.04 mg/g	Langmuir	Pseudo-second order	[126]
Cystene–montmorillonite nanocomposite	Pb (II)	–	–	0.180 mg/g	–	–	[127]
Cloisite–polycaprolactone nanocomposite	Pb (II)	–	–	88%	–	–	[128]
Attapulgite/poly(acrylic acid)	Pb (II)	–	5.0	38.0 mg/g	Freundlich	Pseudo-second order	[129]

## Data Availability

Data sharing is not applicable.

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
