# Peer review of "Clay-Polymer Nanocomposites: Preparations and Utilization for Pollutants Removal"

_materials, 2021, doi:10.3390/ma14061365_

Round 1

Reviewer 1 Report

This is a review paper on clay-polymer nanocomposites. The authors should use the classification of clay minerals and the clay mineral nomenclature as explained in the Handbook of Clay Science (Bergaya et al., Elsevier, first or second edition). This consistent use facilitates eading and understanding the paper. See especially lines 98-100.

line 106: medium pH change? The CEC consists of a pH-independent part, due to isomorphous substitution and a pH-dependent part, due to terminal OH gropups.

The distinction between exfoliated, intercalated and phase separated clay polymer nanocomposites is not clear.

lines 269-275: removal mechanisms includes surface charge, flow conditions..: these are not mechanisms.

lines 2888-289: powders and tablets are prepared by thin coating...: I do not quite understand

line 383: chemisorption is a chemical reaction between the surface and the dye: which one?

line 441: copper complex mesopores: meaning?

line 463 metallic copper ion: metallic means Cu(O)

Overall, the English must be improved

Author Response

Reviewer 1:

This is a review paper on clay-polymer nanocomposites. The authors should use the classification of clay minerals and the clay mineral nomenclature as explained in the Handbook of Clay Science (Bergaya et al., Elsevier, first or second edition). This consistent use facilitates eading and understanding the paper. See especially lines 98-100.

Statement was added to clarify the classification of clay minerals (Lines 108-112). A table was also added (Line 120).

line 106: medium pH change? The CEC consists of a pH-independent part, due to isomorphous substitution and a pH-dependent part, due to terminal OH gropups.

" medium pH change " was omitted from the test. Yes, the process of replacing one structural cation for another of similar size is referred to as isomorphous substitution. This isomorphous represents the primary source of both negative and positive charges in clay minerals

The distinction between exfoliated, intercalated and phase separated clay polymer nanocomposites is not clear.

In the exfoliated nanocomposites, the clay layers are fully dispersed within the polymer matrix. In intercalated nanocomposites, the clay layers are inserted within the polymer matrix (Line 152-155).

Lines 269-275: removal mechanisms includes surface charge, flow conditions..: these are not mechanisms.

The statement was corrected

"The removal of microbes using clay-polymers NCs is controlled by various factors such as the properties of the adsorbent surface (surface charge, composition, etc.), the characteristics of the bacteria and the operating conditions (pH, temperature, dosage, etc.)" (Lines 332-335).

Lines 288-289: powders and tablets are prepared by thin coating...: I do not quite understand

The statement was corrected

"Several clay-polymers NCs in the form of powder or tablets were prepared using the thin-coated layer methods. These methods provide an enhancement of the adsorbent reusability after adsorption of toxic pollutants"(Lines 350-352).

Line 383: chemisorption is a chemical reaction between the surface and the dye: which one?

The dye is congo red. "Chemisorption and electrostatic attractions are the suggested mechanisms of the congo red onto the chitosan/montmorillonite nanocomposite" This statement was added in the text (Line 444-445)

Line 441: copper complex mesopores: meaning?

Mesopores that mean the pores (with diameters between 2 and 50 nm) that exist in the composites. The statement was added in the text (Line 507).

Line 463 metallic copper ion: metallic means Cu(O)

"metallic copper ion" was replaced by Cu(II) ions  

Overall, the English must be improved

The paper was revised and the English was improved

Reviewer 2 Report

Although the topic is interesting, the review seems to be only an enumeration of information/ articles/ papers. This issue should be corrected by connecting/ merging the information presented in order to offer an organized synthesis of the current state, according to an original perspective. The context needs to be more organized and unitary.

Introduction. This section should explain the motivation, defines the topic, provides an appropriate context for reviewing the literature and states the scope of review. The authors should revise the introduction section accordingly.

Main body. The information should be connected/ presented in order to offer a more organized perspective on the synthesis of the current state.

Conclusions. This section should be considerably improved. The conclusions should recapitulate the most important aspects of the existing literature data, assess the current state of the studied literature data, classify important flaws or gaps in current knowledge and outline areas for future study.

References. There is a tendency of self-citation. This should be reconsidered.

I consider that the review needs a carefully, major revision before resubmitting. If the manuscript will not be considerable improved, I will not recommend its publication.

Author Response

Reviewer 2:

Although the topic is interesting, the review seems to be only an enumeration of information/ articles/ papers. This issue should be corrected by connecting/ merging the information presented in order to offer an organized synthesis of the current state, according to an original perspective. The context needs to be more organized and unitary.

  1. Introduction. This section should explain the motivation, defines the topic, provides an appropriate context for reviewing the literature and states the scope of review. The authors should revise the introduction section accordingly.

The introduction was modified as requested by the reviewer. Now the topic is well introduced (Line 83-105)

  1. Main body. The information should be connected/ presented in order to offer a more organized perspective on the synthesis of the current state.

Paragraphs were added in order to connect the information between sections (Line 141-145; Lines 287-280; 293-296)

  1. Conclusions. This section should be considerably improved. The conclusions should recapitulate the most important aspects of the existing literature data, assess the current state of the studied literature data, classify important flaws or gaps in current knowledge and outline areas for future study.

The conclusion was arranged as requested by the reviewer. (Lines 679-)

  1. References. There is a tendency of self-citation. This should be reconsidered.

Some references were replaced by others

  1. I consider that the review needs a carefully, major revision before resubmitting. If the manuscript will not be considerable improved, I will not recommend its publication.

The manuscript was carefully revised and improved.

Reviewer 3 Report

The manuscript entitled: “ Clay-polymer nanocomposites: preparations and utilization for pollutants removal” is relevant for the Materials journal. It based on literature review. It quite well reported the stat-of-art, but there is lack of the authors own opinion and new added value (for example comparative study of different aspects). Additionally, the article requires some minor changes:

  • Authors: the name of country should be the same (1,3 and 4);
  • Abstract: please give the main findings;
  • Introduction: line 51, remove additional bracket;
  • Introduction: please stress why the authors decided on this particular topic (lines: 94-96);
  • The lack of methodology for review; how the articles have been selected, what kind of the databased have been used?
  • Structures and types of clay minerals: the table (or figure) for summarizing the presented knowledge in this chapter should be considered;
  • Nanocomposites of clay-polymers: line 128-129, could you connect this classification with sub chapters? The arrangement of sub-chapters is not proper. Some of them are only 3 sentences. They should be enriched at additional information or rearranged;
  • Biological pollutants removal: Line 235-236, please use italic for the names of bacteria (all chapter);
  • Factors making clay-polymers nanocomposites promising materials for wastewater treatment: please compare the efficiency of different composites.

Author Response

The manuscript entitled: “ Clay-polymer nanocomposites: preparations and utilization for pollutants removal” is relevant for the Materials journal. It based on literature review. It quite well reported the stat-of-art, but there is lack of the authors own opinion and new added value (for example comparative study of different aspects). Additionally, the article requires some minor changes:

  1. Authors: the name of country should be the same (1,3 and 4);

Country name was corrected.

  1. Abstract: please give the main findings;

The abstract was improved

  1. Introduction: line 51, remove additional bracket;

Bracket was eliminated

  1. Introduction: please stress why the authors decided on this particular topic (lines: 94-96);

The introduction was modified as requested by the reviewer. Now the topic is well introduced (Line 83-105)

  1. The lack of methodology for review; how the articles have been selected, what kind of the databased have been used?

The paper is a review the data were collected based on recent publications. Recent results were discussed in this paper

  1. Structures and types of clay minerals: the table (or figure) for summarizing the presented knowledge in this chapter should be considered;

A table summarizing the types and structure of clay minerals was added (Line 120)

  1. Nanocomposites of clay-polymers: line 128-129, could you connect this classification with sub chapters? The arrangement of sub-chapters is not proper. Some of them are only 3 sentences. They should be enriched at additional information or rearranged;

A connection paragraph was added at the end of the section 2. (Lines 141-147)

The sub-sections 3.2 to 3.9 were enriched with additional information (Lines 209-213; 221-229; 236-243; 263-256; 262-263; 272-276; 282-284) .

  1. Biological pollutants removal: Line 235-236, please use italic for the names of bacteria (all chapter);

Names of bacteria were rewritten in italic

  1. Factors making clay-polymers nanocomposites promising materials for wastewater treatment: please compare the efficiency of different composites.

Paragraph comparing the efficiency between clay and composites was added in the section (Lines 632-651)

Round 2

Reviewer 1 Report

This revised paper still has 2 deficiencies: (1) the language is poor; (2) the authors do not use the correct nomenclature of clay mineral science.  I give examples of the second point.

There is a difference between clays and clay minerals. The authors use clay minerals. So they have to avoid the use of clay. If the name of a clay mineral is used, it must not be followed by clay. Example: montmorillonite not montmorillonite clay. Clay minerals consist of layers not platelets.line 114 replace "layers" by "sheets". line 116 and 119: write "sheets" after "octahedral". line 126: replace "layer" by "sheet"; line 157: what are "main traditional microcomposites"? line 168: what is meant by "pure one"? lines 232-234: I do not understand. lines 281-282 are not clear. line 320: what is "quaternary ammonium ether cationic monomer"? lines 468-469 need explanation. line 532: this sentence is too vague.

Author Response

-This revised paper still has 2 deficiencies: (1) the language is poor; (2) the authors do not use the correct nomenclature of clay mineral science.  I give examples of the second point.

- The language was carefully revised

In This review clay minerals was classified as reported by  Lee and Tiwari (2012) and Konta (1995). The same classification was reported recently by Mukhopadhyay et al. (2020)

Lee, S.M.; Tiwari, D. Organo and inorgano-organo-modifid clays in the remediation of aqueous solutions: an overview. Appl. Clay Sci. 2012, 59–60, 84–102.

Konta, J. Clay and man: clay raw materials in the service of man, Appl. Clay Sci. 1995, 10, 275-335.

Mukhopadhyay et al. (2020) Clay–polymer nanocomposites: Progress and challenges for use in sustainable water treatment, Journal of Hazardous Materials, 383, 121125

-There is a difference between clays and clay minerals. The authors use clay minerals. So they have to avoid the use of clay. If the name of a clay mineral is used, it must not be followed by clay. Example: montmorillonite not montmorillonite clay. Clay minerals consist of layers not platelets.

The word "clay" was eliminated as requested

-line 114 replace "layers" by "sheets". line 116 and 119: write "sheets" after "octahedral". line 126: replace "layer" by "sheet";

The requested corrections were conducted

-line 157: what are "main traditional microcomposites"?

the word "traditional " was replaced by "conventional"

line 168: what is meant by "pure one"?

The statement was corrected "pure one" is replaced by "pure oleic acid coated magnetic nanoparticles component" (Line 167)

-lines 232-234: I do not understand.

The paragraph was rewritten: "Polystyrene-clay composites were produced using the addition polymerization technique [34]. A vinyl monomer-montmorillonite intercalate was synthesized by the cation exchange process. Then, free-radical polymerization of styrene in the presence of vinyl monomer-montmorillonite intercalate was achieved leading to the production of polystyrene-montmorillonite materials. The salvation of vinyl monomer-montmorillonite intercalate using acetonitrile enhance the intercalation of styrene between the vinyl monomer- montmorillonite [34]. " (Lines 229-236)

-lines 281-282 are not clear.

The paragraph was rewritten: "The polymerization process of polyolisocyanate precursor–organoclay allowed the production of NCs nanocomposites having an intercalated clay phase (with 50 Å basal spaces) intercalated in the cross-linked polyurethane network. "(Lines 277-279)

-line 320: what is "quaternary ammonium ether cationic monomer"?

The statement was corrected : "the bacterial removal was achieved due to the presence of cationic monomers on clay surface""(Line 313)

-Lines 468-469 need explanation.

The paragraph was rewritten: "In another work, MB was removed from water using hybrid membranes based on kaolin and polystyrene. These membranes were fired at 1000 °C allowing the modification of polystyrene and the formation cavities inside the clay matrix. The resulted materials have porous structure with expanded surface area. An optimum removal of MB from water was obtained with clay membrane loaded with 20% (weight / weight) polystyrene [78]. ""(Lines 433-437)

Line 532: this sentence is too vague.

More information was added "chitosan/attapulgite composites were synthesized and tested for the removal of Cr(III) and Fe(III) from aqueous solution. The maximum adsorption capacity  was 11.65 and 10.41 mg/g for Cr(III) and Fe(III), respectively. These values were obtained with adsorbent dosage of 0.2 g/l…. "   "(Lines 503-509)

Reviewer 2 Report

I consider that the article can be accepted for publication only after a major revision, considering that the authors did not entirely responded to my previous recommendations.

Main body. The information should be connected/ presented in order to offer a more organized perspective on the synthesis of the current state, while keeping only the relevant information. Table 3 is too large. In this form, the review appears too long, making difficult the highlighting of relevant information. 

References. There is a tendency of self-citation (ex. Tahoon). This should be reconsidered.

Author Response

Main body. The information should be connected/ presented in order to offer a more organized perspective on the synthesis of the current state, while keeping only the relevant information. Table 3 is too large. In this form, the review appears too long, making difficult the highlighting of relevant information. 

We improved the paper. All the modifications included in the revised version are highlighted in red.

The size of table 3 was reduced

- References. There is a tendency of self-citation (ex. Tahoon). This should be reconsidered.

References were reconsidered